# Regulation of Protein Transport Pathways by the Cytosolic Hsp90s

**DOI:** 10.3390/biom12081077

**Published:** 2022-08-05

**Authors:** Anna G. Mankovich, Brian C. Freeman

**Affiliations:** School of Molecular and Cellular Biology, University of Illinois, Urbana-Champaign, Urbana, IL 61801, USA

**Keywords:** Hsp90, molecular chaperone, protein transport

## Abstract

The highly conserved molecular chaperone heat shock protein 90 (Hsp90) is well-known for maintaining metastable proteins and mediating various aspects of intracellular protein dynamics. Intriguingly, high-throughput interactome studies suggest that Hsp90 is associated with a variety of other pathways. Here, we will highlight the potential impact of Hsp90 in protein transport. Currently, a limited number of studies have defined a few mechanistic contributions of Hsp90 to protein transport, yet the relevance of hundreds of additional connections between Hsp90 and factors known to aide this process remains unresolved. These interactors broadly support transport pathways including endocytic and exocytic vesicular transport, the transfer of polypeptides across membranes, or unconventional protein secretion. In resolving how Hsp90 contributes to the protein transport process, new therapeutic targets will likely be obtained for the treatment of numerous human health issues, including bacterial infection, cancer metastasis, and neurodegeneration.

## 1. Introduction

For decades, molecular chaperones have been recognized as essential agents in the maintenance of protein homeostasis (proteostasis). One of the earliest-identified molecular chaperones is the highly conserved heat shock protein 90 (Hsp90) [1,2]. Apart from Archaea, Hsp90 is present in almost all bacterial and eukaryotic life [3,4]. In general, the number of Hsp90 homologues expands in parallel with increased cellular complexity [4,5]. For example, the majority of bacteria only contain one nonessential Hsp90 homologue often referred to as high-temperature protein G (HtpG) while eukaryotes can have multiple Hsp90 proteins including several cytoplasmic/nucleoplasmic Hsp90s, mitochondrial TRAP1 (tumor necrosis factor receptor-associated protein 1), endoplasmic reticulum (ER) GRP94 (94 kDa glucose-regulated protein), and chloroplast HSP90C (plastid heat shock protein 90) [5]. Each of the homologues, except for cytoplasmic/nucleoplasmic Hsp90s, is retained in its respective organelle to help maintain protein quality control within that compartment [6,7,8]. Here, we will focus on how the cytoplasmic/nucleoplasmic Hsp90 proteins contribute to the protein transport process.

Although compartmentalization has many benefits, it does require a multi-faceted delivery mechanism to transport biological molecules into, out of, within, and between organelles. Despite being historically implicated as a signaling pathway regulator of steroid hormone receptors and kinases [9,10], Hsp90 is increasingly associated with many other cellular processes, including protein transport [11,12,13]. Given the physiological relevance of protein transport to health and disease, including implications in neurotransmitter release, cell differentiation, bacterial infection, and autophagy [14,15,16,17], it is important to better understand how and when Hsp90 contributes to this process.

To gain insights into how one of the most abundant proteins in a cell cytoplasm/nucleoplasm contributes to life, multiple studies have attempted to identify the physical, genetic, and chemical–genetic interactors of Hsp90 using a variety of unbiased high-throughput screens including two-hybrid, synthetic genetic arrays and mass spectrometry-based tactics [11,12,13,18,19]. Significantly, within the conundrum of hits, there are numerous players with established roles in the transport and secretion pathways [11,12,13]. These connections include proteins involved in various aspects of exocytosis and endocytosis [13]. Focusing on the physical and genetic interactors of budding yeast’s only two isomers, cytoplasmic/nucleoplasmic Hsc82 and Hsp82, Hsp90 is linked to 202 different proteins driving most aspects of intracellular transport and secretion (Figure 1). Hence, Hsp90 likely has a significant influence on the transport process that goes well beyond our current understanding. Perhaps of note, the human homologs of many of the yeast Hsp90-interactors are associated with various diseases, including bacterial infections (e.g., tetanus (SEC18/NSF), infant botulism (YKT6/YKT6), and diphtheria (RRT2/DPH7)), cancer (e.g., breast mucinous carcinoma (TRX2/TXN), primary bone cancer (SLT2/MAPK7), and endometrial cancer (PKH1/PDPK1)), and neurodegeneration (amyotrophic lateral sclerosis (VPS21/RAB5A, CDC48/VCP, VPS60/CHMP5), Parkinson’s disease (BET4/RABGGTA, VPS35/VPS35, RIC1/RIC1), and dementia (CDC48/VCP, VPS60/CHMP5)) [20]. Unfortunately, for the majority of the connections, the molecular/physiological basis for the interaction has not been revealed. In this review, we will highlight the few established roles of Hsp90 in transport as well as underscore areas linked to Hsp90 through a variety of high-throughput screens [11,12,13,18,19].

## 2. Hsp90 and General Principles of Protein Transport

The physical flow of proteins among cellular compartments is a highly proteostasis-dependent process. Depending upon the precise transport step, the contribution from the proteostasis system will vary, including protein unfolding (even partial) to allow transfer, the maintenance of unfolded clients during transfer, polypeptide refolding after transfer, assembling large macromolecular complexes (e.g., vesicle formation), disassembling protein structures (e.g., vesicle fusion), monitoring the health of the transport machinery itself, and mediating the removal of damaged factors including clients or machinery components. If or how Hsp90 might contribute to these or other transport steps is an open investigation. To better understand where Hsp90 might contribute, we will briefly review the primary pathways used in the transport process, and then we will discuss established contributions of Hsp90 relative to protein transport.

Two major intracellular trafficking mechanisms are the endocytic and exocytic pathways [22]. Notably, both mechanisms rely heavily on vesicles, and Hsp90 has been linked to 106 different factors governing vesicle transport, including coat proteins, Rabs, soluble N-ethylmaleimide-sensitive factor attachment protein receptors (SNAREs), and Golgi complex proteins (Figure 1). The endocytic pathway allows for the internalization, recycling, and modification of membrane bound surface proteins such as signaling receptors as well as other cargo from the environment. Significantly, Hsp90 shares a total of 96 linkages to endocytosis and endosomal transport, including vacuolar protein sorting proteins, sorting nexin family proteins, and actin (Figure 1). Proteins are endocytosed through a variety of mechanisms whereby the plasma membrane invaginates to form a vesicle prior to being delivered to an early endosome [23]. At the early endosome, an initial decision is made to either recycle membrane proteins, such as receptors, back to the plasma membrane via recycling endosomes to direct proteins to the trans-Golgi network via the retromer, or to degrade proteins via the lysosome. To accomplish this sorting, cargo leaves the early endosome in intraluminal vesicles to become multivesicular bodies, endosomal carrier vesicles, or late endosomes, which can be sorted into lysosomes or fuse with autophagosomes [24,25,26,27] (Figure 2).

The exocytic or secretory pathway is responsible for the synthesis, folding, modification, and trafficking of proteins that are members of the endomembrane system or destined for secretion. In exocytosis, newly synthesized proteins are inserted into the ER where they undergo maturation through folding and modifications, such as glycosylation and disulfide bond formation [39,40]. Across the exocytosis pathway, Hsp90 is connected to 23 different proteins comprising some myosins, Rabs, and exocyst complex components (Figure 1). From the ER, cargo is transported to the ER-Golgi Intermediate Compartments (ERGICs) via vesicles and then to the cis-Golgi where ER resident proteins are returned to the ER via retrograde transport [41]. In the Golgi, proteins undergo additional carbohydrate modifications and proteolytic processing as the cargo travels from the cis- to trans-Golgi either by vesicular transport or cisternal maturation [39,41]. From the trans-Golgi, cargo vesicles are trafficked to the plasma membrane where they fuse to release secretory proteins or deliver membrane proteins [41] (Figure 2). Hsp90 has been linked to 23 different proteins mediating these late secretory events including syntaxin and vesicle-associated membrane proteins (Figure 1). Furthermore, Hsp90 interacts with an additional 41 proteins that are known to regulate the overall protein transport process comprising Rho GTPase, mitogen-activated protein kinase (MAPK), cyclin-dependent kinase (CDK) and E3 ubiquitin–protein ligase (Figure 1). Although endocytic and exocytic transport mechanisms are well-studied pathways essential to the production, processing, and trafficking of proteins in cells, there is still much to be learned about the underlying mechanisms driving these events, including the contributions of the Hsp90 molecular chaperone.

## 3. Hsp90 Input with Mitochondria and Chloroplast Protein Import

The transport of proteins into mitochondria or chloroplasts requires the assistance of molecular chaperones. As ~95% of mitochondrial and chloroplast proteins are encoded by the nuclear genome and translated by cytoplasmic ribosomes, the post-translational import of proteins is considerable [42]. To successfully transport the various preproteins into these organelles, chaperones, including Hsp90, have been shown to deliver both mitochondrial and chloroplast preproteins to the respective outer membrane translocases [43,44]. In the case of mitochondrial import, Hsp90 works with Hsp70 to target and transport unfolded or hydrophobic preproteins to the translocase of the mitochondrial outer membrane (TOM) complex for import [43,45]. Specifically, Hsp90 docks onto a peripherally associated receptor subunit, Tom70, releasing the preprotein in an ATP-dependent manner either to be bound first by Tom70 or directly to the TOM pore complex [45,46]. To facilitate chloroplast import, Hsp90 binds precursor proteins and interacts with Toc64, a receptor subunit of the translocon of the outer envelope of chloroplasts (TOC) complex [44]. Similarly to mitochondrial transfer, the import of preproteins into the chloroplasts is Hsp90- and ATP-dependent [44] (Figure 3A,B).

Interestingly, Hsp90 fosters transfer across either the organelle’s outer membrane by docking onto a receptor (Toc64 for chloroplasts or Tom70 for mitochondria) using a similar clamp-type tetratricopeptide repeat (TPR) domain present in the receptors [43,44,47]. At least within chloroplasts, client proteins can remain reliant on an Hsp90 homolog after crossing the outer membrane. Within the stroma, HSP90C and chloroplast Hsp90 (cpHsp70) and Hsp93 aid the translocation of proteins across the inner chloroplast membrane while Hsp90C also targets thylakoid lumen proteins to the thylakoid membrane translocase SecY1 [8,48,49,50] (Figure 3A,B). Hsp90’s common role in preprotein targeting and transport to endosymbiotically derived organelles may indicate that in eukaryotic evolution, chaperones were used as an effective solution to import preproteins translated in the cytosol [42,47].

**Figure 3 biomolecules-12-01077-f003:**
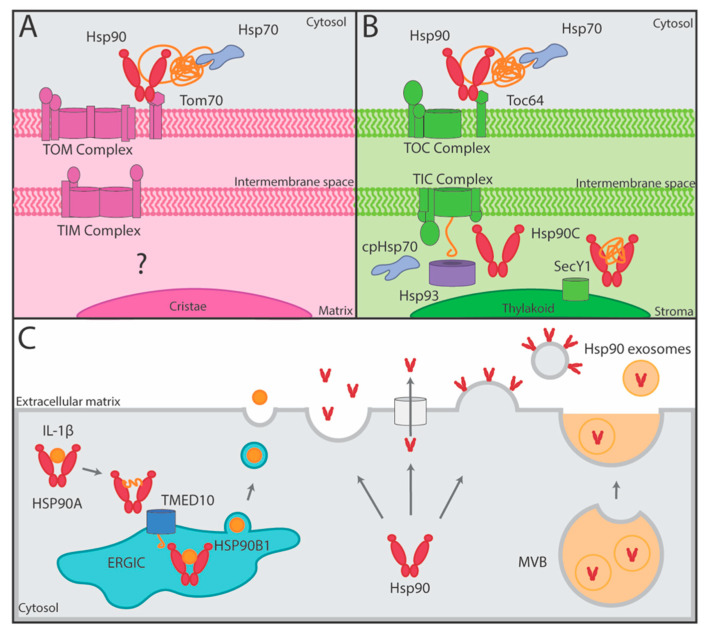
The role of Hsp90 in mitochondrial and chloroplast import as well as unconventional protein secretion (UPS). (**A**) Hsp90 with the aid of Hsp70 delivers preproteins to the TOM complex in the outer membrane of mitochondria through interactions with the TPR domain (hexagon) of peripherally associated Tom70 [43,45]. (**B**) Hsp90 with the aid of Hsp70 delivers preproteins to the TOC complex in the outer membrane of chloroplasts through interactions with the TPR domain (hexagon) of peripherally associated Toc64 [44]. Chaperones in the chloroplast stroma including cpHsp70, Hsp93, and Hsp90C aid in protein transport through the TIC complex in the inner membrane and Hsp90C additionally targets proteins to the SecY1 translocase in the thylakoid membrane [8,48,49,50]. (**C**) HSP90A mediates UPS of IL-1β by unfolding the protein to allow for its transport into the ERGIC through the TMED10 translocase aided by HSP0B1, and then from ERGIC, IL-1β is secreted in vesicles (left) [51]. eHsp90 is thought to be secreted by UPS through vesicle fusion, across membrane transporters, on the surface of exosomes, and/or fusion of MVBs with the plasma membrane (right) [52,53].

## 4. Hsp90 and Unconventional Protein Secretion

Classically, secreted proteins have a signal peptide or leader sequence that targets the polypeptides for ER–Golgi trafficking via vesicles to eventually export the factors out of a cell [54,55]. Still, there are secreted proteins that lack such sequences and bypass the ER–Golgi route. These “leaderless” proteins are exported by unconventional protein secretion (UPS) that uses both non-vesicular and vesicular paths to export proteins [56]. One mechanism, which is dependent upon Hsp90, is the secretion of interleukin-1β (IL-1β) by mammalian cells during autophagy known as the TMED10-channeled UPS (THU) [17,51]. Following autophagy induction, IL-1β is produced and captured in its mature form by cytosolic HSP90A, potentially though the binding of KFERQ-like motifs in IL-1β [17,51]. It has been suggested that HSP90A then unfolds IL-1β to expose a signal motif allowing direct translocations into the ERGIC through transmembrane Emp24 domain-containing protein 10 (TMED10) [51] (Figure 3C). Of note, the import of IL-1β by TMED10 is also aided by HSP90B1, although HSP90B1 mechanistic contributions are less well understood [51]. The idea that Hsp90 recognizes Hsp90 through its KFERQ-like motifs is questionable, however, because so far only Hsc70, an Hsp70 family member, has been shown to bind KFERQ motifs, which are primarily used to target proteins to lysosomes for degradation in Chaperone-Mediated Autophagy (CMA) [57]. In CMA, Hsp90 plays a more indirect role through the stabilization and oligomerization of lysosome-associated membrane protein type 2A (LAMP2A) from inside the lysosome to allow for the Hsc70-delivered proteins to enter the lysosome through the LAMP2A translocation complex [58,59]. Thus, it is possible that Hsp90 may recognize IL-1β for UPS through another mechanism other than its KFERQ-like motifs.

Intriguingly, many heat shock proteins including Hsp60, Hsp27, Hsp20, Hsp70, and Hsp90 itself are likely exported out of cells via UPS [60]. Migratory cells during wound healing or cancerous growth continuously secrete Hsp90 [61,62,63] and normal cells exposed to a variety of stresses, including heat, hypoxia, serum starvation, reactive oxygen, or virus infection, transiently export Hsp90 [62,64,65,66,67,68]. Minimally, extracellular Hsp90 (eHsp90) increases cell motility. In the case of wound healing, Hsp90 is secreted in response to hypoxia at the wound site where Hypoxia-Inducible Factor-1alpha (HIF-1α) promotes the secretion of Hsp90α [62]. Notably, eHsp90 accelerates wound healing by inducing the migration of dermal fibroblasts, including facilitating wound healing in mice when applied topically [62,69].

In contrast to the beneficial role in wound healing, the secretion of Hsp90 on its own or on the outer surface of vesicles has been shown to increase cancer metastasis [70]. The eHsp90 enhances metastasis by dysregulating the extracellular matrix (ECM) through the activation of ECM-modifying proteases [52,71]. Thus, there is growing interest in utilizing eHsp90 as a biomarker and/or target for cancer treatment [52]. Despite the potential importance of eHsp90, the mechanism by which Hsp90 is secreted from cells is not fully understood. Minimally, Hsp90 can exit a cell via exosomes generated in the endocytic pathway rather than the canonical secretory pathway [72] (Figure 3C). How Hsp90 is loaded into these exosomes prior to export is not clear [52]. Nevertheless, Hsp90’s influence with other types of vesicles has been shown.

## 5. Influence of Hsp90 in Endocytic Vesicle Transport

Hsp90 has been implicated in several endocytic mechanisms including endosome vesicle transport and recycling. A well-studied impact of Hsp90 is on the regulation of Rab recycling—Rabs are members of the Ras GTPase family [73]. Rab-GTPases generally regulate the assembly and disassembly of complexes that enable vesicle targeting and fusion [30]. Following the activation of Rab by GTPase-activating proteins (GAPs), GDP-bound Rab is retrieved from membranes by GDP-dissociation inhibitors (GDIs), thus enabling continuous vesicle transport [74,75]. Hsp90 modulates this recycling process by binding to GDI and Rab. Hsp90 recruits GDI to the membrane and configures GDI into an open confirmation to bind the geranylgeranyl (GG) lipids anchoring Rab-GDP to the membrane, triggering its release into the cytosol [73]. Hsp90 has been shown to interact with Rab11b in osteoclasts to mediate the transport of macrophage colony-stimulating factor receptor (c-fms) and receptor activator of nuclear factor kappa B (RANK) surface receptors in early to late endosomes and also to lysosomes for degradation [15] (Figure 2). This Hsp90-mediated endosomal transport of receptors to lysosomes allows for the proper regulation of osteoclastogenesis and the differentiation of hematopoietic precursors to osteoclasts that are critical for bone homeostasis [15,76].

The endocytic pathway mediated by Rab11 and Hsp90 can be hijacked by *Neisseria meningitidis* to aid in bacterial internalization [36]. In this study, endocytic vesicles containing both *Neisseria meningitidis* adhesin A (NadA) protein and Hsp90 recruited Rab11 in human epithelial cells, causing the NadA endosomes to be recycled to the cell’s surface. Using the membrane-impermeable Hsp90 inhibitor FITC-GA to selectively inhibit the eHsp90 prevented the recruitment of Rab11 and subsequent endosomal recycling [36]. Hsp90 has been shown to bind NadA and interfere with bacterial attachment [16,36]. Hence, eHsp90 might serve as an interesting target to combat bacterial infection for agents entering through an endosomal transport mechanism. Significantly, disruption of membrane proteins recycling upon Hsp90 inhibition also has been observed with the cancer-associated ErbB2 tyrosine kinase receptor [37,77,78]. Normally ErbB2 is trafficked in early endosomes for recycling; however, upon treatment with an Hsp90 inhibitor, these receptors are instead routed to multivesicular endosomes and lysosomal compartments [37]. These compartments were found to have a modified ultrastructure, which is more tubular than under normal conditions [37]. It is speculated that this disruption of normal transport and structure is due to Hsp90’s interactions with Rab12, which normally localizes to early/recycling endosomes and lysosomes or through Hsp90’s regulation of cytoskeletal dynamics [37,78].

## 6. Modulation of Exocytic Intracellular Transport by Hsp90

In addition to endocytic events, Hsp90 aids exocytic pathways, including ER-Golgi vesicular transport and protein secretion. The influence of Hsp90 on these events has been primarily delineated by tracking the transport of vesicular stomatitis virus glycoprotein (VSV-G) in mammalian cells [31,32,73]. In these studies, the loss of Hsp90 blocks ER to Golgi and intra-Golgi transport, as evidenced by an impaired anterograde vesicle transport and Golgi fragmentation [31,32]. Of note, these observations have been attributed to multiple Hsp90 roles, including the following: (1) Hsp90’s regulation of vesicular transport by associating with the membrane-bound protein VAPA in complex with the co-chaperone tetratricopeptide repeat protein TTC1 [31]; (2) Hsp90’s modulation of microtubule stability by controlling microtubule-associated protein 4 (MAP4), which is essential for maintaining microtubule acetylation and stability [32]; and (3) Hsp90’s control of Rab1 recycling. Rab1 is responsible for ER to Golgi trafficking with mammalian Rab1b also dictating the transport of proteins through the cis- and medial-Golgi compartments [28,29,30] (Figure 2). Hence, Hsp90 facilitates ER to Golgi vesicular transport by both promoting vesicle targeting and maintaining the structure of the Golgi.

In the secretory pathway Hsp90 has been shown to have diverse functions with exosome release. For instance, Hsp90 influences membrane conformation to promote the fusion of multivesicular bodies and the plasma membrane [38] (Figure 2). This membrane-remodeling activity is dependent upon an evolutionarily conserved amphipathic helix in Hsp90 both in vitro and in vivo at synapses and is further promoted by the Hsp90 cochaperone HOP [38]. Besides HOP, the AHA1 cochaperone fosters the release of secretory vesicles associated with Rab3, which supports the cell migration of cancer cells [33]. The involvement of Hsp90 with Rab3 is a common thread in many exocytosis events (Figure 2). For example, Hsp90 is key to Ca^2+^-triggered neurotransmitter release through the αGDI-dependent recycling of Rab3A [34]. Similarly to other Rabs, Hsp90 forms a complex with αGDI to remove Rab3A from the lipid bilayer during neurotransmitter release [34]. Significantly, the Hsp90-αGDI regulation of Rab3A controls the association of α-Synuclein, a presynaptic protein linked to neurodegeneration, with the synaptic membrane [35]. At the pre-synaptic membrane, α-synuclein associates preferentially with Rab3A-GTP and is subsequently released from the membrane following the actions of GDI and Hsp90 [35]. Of note, the Hsp90-GDI regulation of Rab11A, which is typically involved in the recycling of endosomes, has been linked to the secretion of α-synuclein, and the Hsp90-dependent release of α-synuclein is associated with increased neurotoxicity [79]. Although not specifically connected to Rab recycling, Hsp90 has also been found to mediate the transport of Aldo-Keto Reductase 1B10 (AKR1B10), a tumor biomarker, by regulating its transport to lysosomes or secretion out of the cell [80]. Hence, an improved understanding of how Hsp90 governs secretion may lead to improved future therapies for treating cancer and neurodegeneration.

## 7. Conclusions

The cytoplasmic/nucleoplasmic Hsp90 interactome contains hundreds of connections to factors working in various aspects of the protein-transport process (Figure 1). Yet, the defined contributions of Hsp90 to the various transport pathways remains limited. Nevertheless, it is clear that Hsp90 facilitates central features of protein transport, including promoting endocytic and exocytic vesicular transport, docking clients onto membrane translocation machinery, and fostering unconventional protein secretion [32,43,44,73]. Perhaps significantly, the established roles of Hsp90 in transport have important health implications as the chaperone-dependent steps link to wound healing, bacterial infection, cancer metastasis, and neurodegenerative diseases [16,35,36,62,70,79]. Given the number of connections that have yet to be resolved both mechanistically and physiologically, it is probable that the relevance of the cytoplasmic/nucleoplasmic Hsp90s with protein transports will continue to grow. Beyond these Hsp90 homologues, it is important to consider how organelle Hsp90s (GRP94, TRAP1, and HSP90C) add to the influence of the Hsp90 system on protein transport. For instance, does the dependence of SARS-CoV-2 on Hsp90 [74,75,76,77] relate to its influence on the protein transport process? Minimally, it is apparent that Hsp90’s role in proteostasis will extend well beyond the maintenance of metastable clients.

## Figures and Tables

**Figure 1 biomolecules-12-01077-f001:**
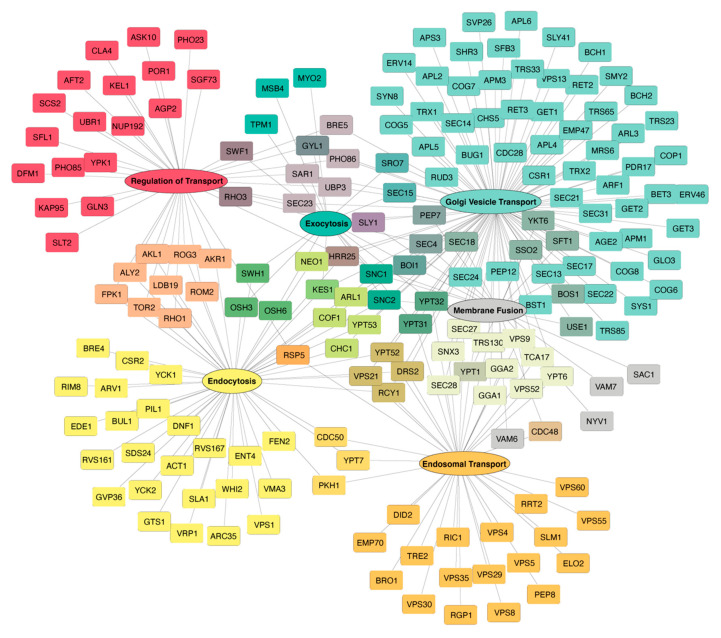
Protein transport factors linked to yeast Hsp90. The transport pathways mediated by the physical and genetic interactors of Hsc82 and Hsp82 were determined using the Gene Ontology Slim Term Mapper from Saccharomyces Genome Database (http://genome-www.stanford.edu/Saccharomyces/ accessed on 6 June 2022). Six Gene Ontology terms relevant to protein transport were selected including Golgi Vesicle Transport, Endocytosis, Endosomal Transport, Regulation of Transport, Exocytosis, and Membrane Fusion. The interactors were organized into the shown interaction map using Cytoscape [21] by setting the Gene Ontology term as a source interactor and each gene as a target interactor.

**Figure 2 biomolecules-12-01077-f002:**
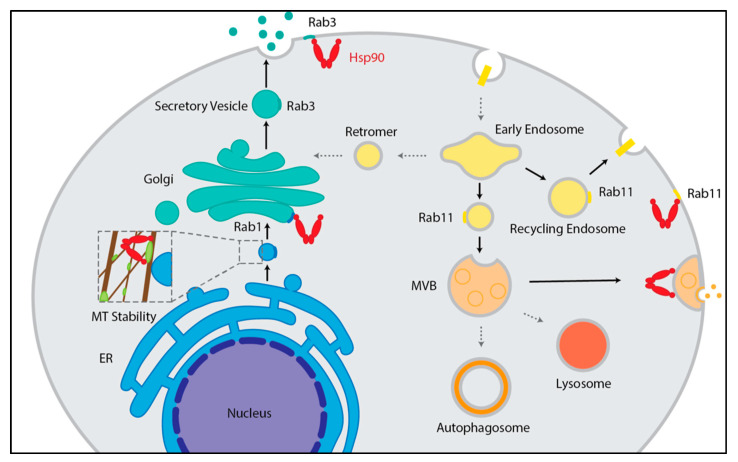
Hsp90 contributions to exocytic and endocytic trafficking. In the exocytic pathway (left), Hsp90 (red) aides in anterograde vesicle trafficking between the ER and Golgi by supporting microtubule stability through the binding of MAP4 (light green) [28,29,30,31,32], and the recycling of Rab1 and Rab3 to enable vesicle targeting and fusion with the Golgi and plasma membrane [28,29,33,34,35]. In the endocytic pathway (right), Hsp90 supports the transport of recycling endosomes as well as transport from early to late endosomes/multivesicular bodies (MVB) through the recycling of Rab11 [15,36,37]. Hsp90 also allows for the fusion of MVB with the plasma membrane by aiding in membrane deformations [38]. Transport events that Hsp90 is involved in are noted by solid black arrows.

## Data Availability

Not applicable.

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
