# Peer review of "Regulation of Protein Transport Pathways by the Cytosolic Hsp90s"

_biomolecules, 2022, doi:10.3390/biom12081077_

Round 1
Reviewer 1 Report
The manuscript entitled "Regulation of Protein Transport Pathways by the Cytosolic 2 Hsp90s" is well argued and explains well the possible roles of HSP90 in intracellular and extracellular pathways. In line 150 authors said: "Still, there are secreted proteins that lack such sequences and bypass the ER-Golgi route. These “leaderless” proteins are exported by Unconventional Protein (UPS) that uses both non-vesicular and vesicular paths to export proteins". Are there other chaperones (for example Hsp60) that are released in an unconventional way? You could add a comment on this in the text.
Author Response
We are very grateful to each reviewer for taking the time to assess our manuscript “Regulation of Protein Transport Pathways by the Cytosolic Hsp90s” and provide their thoughtful and helpful comments and suggestions. Their insight has improved the overall quality and depth of this manuscript. Please find our point-by-point response to all the reviewers’ comments and concerns.
Reviewer #1
The manuscript entitled "Regulation of Protein Transport Pathways by the Cytosolic 2 Hsp90s" is well argued and explains well the possible roles of HSP90 in intracellular and extracellular pathways. In line 150 authors said: "Still, there are secreted proteins that lack such sequences and bypass the ER-Golgi route. These “leaderless” proteins are exported by Unconventional Protein (UPS) that uses both non-vesicular and vesicular paths to export proteins". Are there other chaperones (for example Hsp60) that are released in an unconventional way? You could add a comment on this in the text.
We are thankful to this reviewer for their kind comments.
We added to line 196-197 indicating that other heat shock proteins including Hsp60, Hsp27, Hsp20, and Hsp70 are released extracellularly through UPS.
Reviewer 2 Report
Dear Authors,
I would like to thank you for your manuscript. I think this is a great opportunity to congratulate you on this comprehensive review of how Hsp90's involvement in transport mechanisms contributes to several disease states. Although I highlight the very good quality of the job, there were some minor limitations to the review that may be addressed to improve the quality. The main limitation is in the data validation of the interaction network in Figure 1, It would make it easier to clearly define the nodes and edges in these networks and how Hsp90 is linked to each of these factors are all these direct/indirect linkages. I list my detailed minor comments about the review below.
Limitations:
Line 10, briefly mention these original and the 'additional' pathways to be clear.
line 12, sentence too long.
Lines 18 and 40, The manuscript proposed to highlight how some of the human health issues may be resolved through Hsp90 transport systems but there is not much detail to that in the main body text to justify its inclusion in the abstract. This may be addressed by a brief description of where/how protein transport is relevant in health/disease and what implications it has in neurotransmitter release and other mentioned diseases.
Line 59, figure 1 forms an integral part of this manuscript but the data has not been compared and validated with other interactome databases to validate these findings. In addition, the interactome should be adopted to suit the human Hsp90 homologs as the review is focused on human diseases, not yeast.
Lines 114-115, validate the Hsp90 proteins and comment on the human equivalents.
line 142, mention these factors referred here.
Line 156, mention the corresponding motif on Hsp90 that interacts with the KFERQ if known.
Line 182-191, add a figure to visually explain this process.
Line 210, explain the modifications.
Author Response
We are very grateful to each reviewer for taking the time to assess our manuscript “Regulation of Protein Transport Pathways by the Cytosolic Hsp90s” and provide their thoughtful and helpful comments and suggestions. Their insight has improved the overall quality and depth of this manuscript. Please find our point-by-point response to all the reviewers’ comments and concerns.
Reviewer #2
I would like to thank you for your manuscript. I think this is a great opportunity to congratulate you on this comprehensive review of how Hsp90's involvement in transport mechanisms contributes to several disease states. Although I highlight the very good quality of the job, there were some minor limitations to the review that may be addressed to improve the quality. The main limitation is in the data validation of the interaction network in Figure 1, It would make it easier to clearly define the nodes and edges in these networks and how Hsp90 is linked to each of these factors are all these direct/indirect linkages. I list my detailed minor comments about the review below.
We’d like to thank Reviewer 2 for the supportive comments.
Limitations:
Line 10, briefly mention these original and the 'additional' pathways to be clear.
Although not explicitly laid out in line 10 the “original” pathways that Hsp90 is associated with are mentioned in the previous sentence (line 8-10) “maintaining metastable proteins” and the additional pathways are mentioned in the following sentence “protein transport”. Other pathways have not been included in the abstract as the main focus of the review is on Hsp90’s role in transport instead of a broader discussion of Hsp90’s role beyond maintaining proteostasis.
line 12, sentence too long.
This sentence has been divided and edited to increase clarity.
Lines 18 and 40, The manuscript proposed to highlight how some of the human health issues may be resolved through Hsp90 transport systems but there is not much detail to that in the main body text to justify its inclusion in the abstract. This may be addressed by a brief description of where/how protein transport is relevant in health/disease and what implications it has in neurotransmitter release and other mentioned diseases.
To further highlight how Hsp90’s involvement in protein transport is related to human disease we have included a brief description (Lines 54-62) connecting the human homologues of the Hsp90 linked yeast transport proteins to the diseases they are associated with based on results from a human genome database (GeneCards). We thank the reviewer for helping to increase the interests related to human disease.
Line 59, figure 1 forms an integral part of this manuscript but the data has not been compared and validated with other interactome databases to validate these findings. In addition, the interactome should be adopted to suit the human Hsp90 homologs as the review is focused on human diseases, not yeast.
We are exploiting the published interactome information on the Saccharomyces Genome database since it is the most extensive interaction database that includes genetic interactions, which form a large contingent of the interactome. As this is a literature review, it is unclear what is meant by “validated” since empirical validation would be beyond the scope of this literature review. Furthermore, not all yeast proteins linked to Hsp90 have human homologues therefore we have chosen not to translate figure 1 into human proteins. However, since not all readers might be familiar with the yeast proteins in Figure 1, we have included examples of the types of proteins that were linked to Hsp90 that are also present in humans in the text based on each transport category. These inclusions should give readers a better idea of what kinds of proteins have been linked to Hsp90 and help draw a more direct line to Hsp90’s role in transport in humans. See lines 82-84, 87-88, 110-111, 118-119, and 120-122 for changes.
Lines 114-115, validate the Hsp90 proteins and comment on the human equivalents.
We have included examples of classes of proteins that are also present in humans. See lines 88-90, 93-94, 116-117, 124-125, and 126-128 for changes.
line 142, mention these factors referred here.
The factors referred to in this line are thylakoid lumen proteins that are encoded in the nucleus and are imported into the chloroplast across the outer, inner, and thylakoid membrane. The text has been changed to specify the “factors” as “thylakoid lumen proteins” to avoid confusion. (See line 155)
Line 156, mention the corresponding motif on Hsp90 that interacts with the KFERQ if known.
There is no known motif on Hsp90 that interacts with the KFERQ. A note has been made in the text (lines 185-195) that it is less likely that Hsp90 binds this motif because Hsc70 is primarily found to bind KFERQ especially in the case of chaperone mediated autophagy. This should clarify that although there is the potential for Hsp90 to interact with IL-1β through KFERQ it is more likely that it is through a different recognition mechanism.
Line 182-191, add a figure to visually explain this process.
Figure 1 already includes a visual summary of the Rabs that Hsp90 regulates including representations of which membranes they are localized to. We believe that this figure in addition to the detailed description included in the text is sufficient to explain this process.
Line 210, explain the modifications.
The “modified” multivesicular endosomes and lysosomal compartments are a reference to a change in the observed structure of these cellular compartments upon treatment with an Hsp90 inhibitor. The text has been changed in lines 246-250 to specify that the “modification” is a change to a more tubular structure which is thought be the result of losing Hsp90’s functions in vesicle fusion/targeting and regulation of the cytoskeleton.
Reviewer 3 Report
The manuscript by Mankovich and Freeman provided a comprehensive review on the roles and mechanisms of Hsp90s in various protein transport pathways. The manuscript is well-written and well-organized. It will be an important contribution to the chaperone field.
Two suggestions:
1) Adding figures on the mechanisms of Hsp90s in protein import into mitochondria and chloroplasts and Unconventional Protein secretion. It will make these complicated processes easier to understand.
2) Hsp90 is also involved in protein import into lysosome in chaperone-mediated autophagy. Adding a short description will enhance the breadth of the manuscript.
A minor point:
In Figure 1, it will enhance the clarity if the processes such as Regulation of transport and Endocytosis are in bold.
Author Response
We are very grateful to each reviewer for taking the time to assess our manuscript “Regulation of Protein Transport Pathways by the Cytosolic Hsp90s” and provide their thoughtful and helpful comments and suggestions. Their insight has improved the overall quality and depth of this manuscript. Please find our point-by-point response to all the reviewers’ comments and concerns.
Reviewer #3
The manuscript by Mankovich and Freeman provided a comprehensive review on the roles and mechanisms of Hsp90s in various protein transport pathways. The manuscript is well-written and well-organized. It will be an important contribution to the chaperone field.
We are grateful to this reviewer for their kind words and support.
Two suggestions:
1) Adding figures on the mechanisms of Hsp90s in protein import into mitochondria and chloroplasts and Unconventional Protein secretion. It will make these complicated processes easier to understand.
The suggested figures have been combined into a new figure, Figure 3, on pages 5-6. Protein import into mitochondria and chloroplasts have been visualized side-by side as Figure 3a & 3b in order to highlight the similarities and differences discussed in the text between Hsp90’s role in protein import into these two organelles. Figure 3c demonstrates Hsp90’s role in UPS of the protein IL-1β as well as the UPS of Hsp90 itself into the extracellular space. We believe the inclusion of these figures makes it easier to understand these processes and we thank the reviewer for the suggestion.
2) Hsp90 is also involved in protein import into lysosome in chaperone-mediated autophagy. Adding a short description will enhance the breadth of the manuscript.
A note has been made that Hsp90 is involved in CMA within the section on unconventional protein secretion. This should address another indirect way that Hsp90 mediates protein import though in the context of protein degradation (lines 190-195).
A minor point:
In Figure 1, it will enhance the clarity if the processes such as Regulation of transport and Endocytosis are in bold.
The nodes of the Hsp90 interaction map of protein transport factors in figure 1 have been bolded to make the processes clearer.